# Nationwide Laboratory Surveillance of Progressive Multifocal Leukoencephalopathy in Japan: Fiscal Years 2011–2020

**DOI:** 10.3390/v15040968

**Published:** 2023-04-14

**Authors:** Kazuo Nakamichi, Yoshiharu Miura, Toshio Shimokawa, Kenta Takahashi, Tadaki Suzuki, Nobuaki Funata, Masafumi Harada, Koichiro Mori, Nobuo Sanjo, Motohiro Yukitake, Kazuya Takahashi, Tsuyoshi Hamaguchi, Shoko Izaki, Satoru Oji, Jin Nakahara, Ryusuke Ae, Koki Kosami, Souichi Nukuzuma, Yosikazu Nakamura, Kyoichi Nomura, Shuji Kishida, Hidehiro Mizusawa, Masahito Yamada, Masaki Takao, Hideki Ebihara, Masayuki Saijo

**Affiliations:** 1Department of Virology 1, National Institute of Infectious Diseases, Shinjuku-ku, Tokyo 162-8640, Japan; hebihara@niid.go.jp (H.E.); masayuki.saijo@doc.city.sapporo.jp (M.S.); 2Department of Neurology, Tokyo Metropolitan Cancer and Infectious Diseases Center Komagome Hospital, Bunkyo-ku, Tokyo 113-8677, Japan; yoshiharu_miura@tmhp.jp; 3Department of Medical Data Science, Graduate School of Medicine, Wakayama Medical University, Wakayama 641-8509, Japan; shimokaw@wakayama-med.ac.jp; 4Department of Pathology, National Institute of Infectious Diseases, Shinjuku-ku, Tokyo 162-8640, Japan; tkenta@niid.go.jp (K.T.); tksuzuki@niid.go.jp (T.S.); 5Department of Pathology, Tokyo Metropolitan Cancer and Infectious Diseases Center Komagome Hospital, Bunkyo-ku, Tokyo 113-8677, Japan; nobuaki_funata@tmhp.jp; 6Department of Radiology, Tokushima University School of Medicine, Tokushima 770-8503, Japan; masafumi@tokushima-u.ac.jp; 7Department of Radiology, Tokyo Metropolitan Cancer and Infectious Diseases Center Komagome Hospital, Bunkyo-ku, Tokyo 113-8677, Japan; momosan0929@gmail.com; 8Department of Neurology and Neurological Science, Tokyo Medical and Dental University Graduate School of Medical and Dental Sciences, Bunkyo-ku, Tokyo 113-8510, Japan; n-sanjo.nuro@tmd.ac.jp; 9Department of Neurology, Kouhoukai Takagi Hospital, Okawa-shi 831-0016, Fukuoka, Japan; yukitake@b1.bunbun.ne.jp; 10Department of Neurology, Hokuriku Brain and Neuromuscular Disease Center, National Hospital Organization Iou National Hospital, Kanazawa-shi 920-0192, Ishikawa, Japan; takahashi.kazuya.nx@mail.hosp.go.jp; 11Department of Neurology, Kanazawa Medical University, Kahoku-gun 920-0293, Ishikawa, Japan; gom56@kanazawa-med.ac.jp; 12Department of Neurology, National Hospital Organization Saitama Hospital, Wako-shi 351-0102, Saitama, Japan; izaki.shoko.dn@mail.hosp.go.jp; 13Department of Neurology, Saitama Medical Center, Saitama Medical University, Kawagoe-shi 350-8550, Saitama, Japan; osatoru@saitama-med.ac.jp (S.O.); drno@saitama-med.ac.jp (K.N.); 14Department of Neurology, Keio University School of Medicine, Shinjuku-ku, Tokyo 160-8582, Japan; nakahara@a6.keio.jp; 15Division of Public Health, Center for Community Medicine, Jichi Medical University, Shimotsuke-shi 329-0498, Tochigi, Japan; shirouae@jichi.ac.jp (R.A.); k.kosami@jichi.ac.jp (K.K.); nakamuyk@jichi.ac.jp (Y.N.); 16Department of Infectious Diseases, Kobe Institute of Health, Kobe-shi 650-0046, Hyogo, Japan; s-nuku@gj8.so-net.ne.jp; 17Higashimatsuyama Municipal Hospital, Higashimatsuyama-shi 355-0005, Saitama, Japan; 18Department of Neurology, Narita Tomisato Tokushukai Hospital, Tomisato-shi 286-0201, Chiba, Japan; s-kisida@biscuit.ocn.ne.jp; 19Department of Neurology, National Center Hospital, National Center of Neurology and Psychiatry, Kodaira-shi, Tokyo 187-8551, Japan; mizusawa@ncnp.go.jp; 20Division of Neurology, Department of Internal Medicine, Kudanzaka Hospital, Chiyoda-ku, Tokyo 102-0074, Japan; m-yamada@kudanzaka.com; 21Department of Laboratory Medicine, National Center Hospital, National Center of Neurology and Psychiatry, Kodaira-shi, Tokyo 187-8551, Japan; msktakaobrb@ncnp.go.jp; 22Department of General Internal Medicine, National Center Hospital, National Center of Neurology and Psychiatry, Kodaira-shi, Tokyo 187-8551, Japan; 23Medical Affairs Department, Health and Welfare Bureau, Sapporo-shi 060-0042, Hokkaido, Japan

**Keywords:** cerebrospinal fluid, JC virus, laboratory surveillance, progressive multifocal leukoencephalopathy, real-time PCR testing

## Abstract

Progressive multifocal leukoencephalopathy (PML) is a devastating demyelinating disease caused by JC virus (JCV), predominantly affecting patients with impaired cellular immunity. PML is a non-reportable disease with a few exceptions, making national surveillance difficult. In Japan, polymerase chain reaction (PCR) testing for JCV in the cerebrospinal fluid (CSF) is performed at the National Institute of Infectious Diseases to support PML diagnosis. To clarify the overall profile of PML in Japan, patient data provided at the time of CSF-JCV testing over 10 years (FY2011–2020) were analyzed. PCR testing for 1537 new suspected PML cases was conducted, and 288 (18.7%) patients tested positive for CSF-JCV. An analysis of the clinical information on all individuals tested revealed characteristics of PML cases, including the geographic distribution, age and sex patterns, and CSF-JCV-positivity rates among the study subjects for each type of underlying condition. During the last five years of the study period, a surveillance system utilizing ultrasensitive PCR testing and widespread clinical attention to PML led to the detection of CSF-JCV in the earlier stages of the disease. The results of this study will provide valuable information not only for PML diagnosis, but also for the treatment of PML-predisposing conditions.

## 1. Introduction

Progressive multifocal leukoencephalopathy (PML) is a rare but devastating demyelinating disorder caused by JC virus (JCV). JCV is a relatively small virus with a circular double-stranded DNA genome and is classified in the *Polyomaviridae* family [1,2,3,4]. The International Committee on Taxonomy of Viruses recently adopted the binomial nomenclature *Betapolyomavirus secuhominis* as the species name of this virus [5]. However, the common virus name JC virus is still registered [6]. JCV typically infects individuals during childhood and establishes persistent asymptomatic infection in the kidneys and urinary tract, as well as latent infection in other sites, such as lymph nodes and bone marrow [2,7,8,9,10,11,12]. JCV is widespread in humans, with approximately 60% of healthy adults being serologically positive for this virus [13]. Non-pathogenic JCV with persistent or latent infection has a stable genomic DNA sequence and is referred to as the archetype [14,15].

In individuals with immunocompromised statuses, including those undergoing therapies that affect cellular immunity, JCV can reactivate and replicate in oligodendrocytes, leading to brain demyelination [1,16,17,18,19]. The JCV responsible for causing PML has a hypervariable mutation with deletions and/or duplications in DNA sequences of the non-coding control region (known as the regulatory or transcription control regions) within the viral genome and is referred to as the prototype [9,16,20,21,22,23]. PML occurs in the presence of various immunosuppressive conditions, such as HIV infection, hematologic malignancies, and organ transplantation [2,16,24,25]. In the past decade, there has been an increase in the incidence of PML in patients receiving immunosuppressive or immunomodulatory therapies for autoimmune diseases [1,2,24,26,27].

It is crucial to clarify the clinical features and predisposing conditions of PML to reduce its risk and improve diagnosis. However, in most cases, reporting the occurrence of PML to the government is not mandatory, except for drug-safety monitoring purposes. Despite comprehensive studies on PML with possible adverse events caused by pharmaceutical products [28,29,30,31,32,33,34,35,36,37,38,39], gathering comprehensive information on each PML case across Japan has been challenging. Historically, several strategies have been employed to identify and analyze non-reportable PML cases, including single-center chart reviews of empirical cases [40,41,42,43,44], multicenter observational studies of patients primarily with targeted clinical conditions [25,45,46,47,48,49,50], estimates based on searches of large-scale public databases containing relevant information [51,52,53,54,55,56,57,58], and literature or systematic reviews of individual case reports in related fields [43,47,59,60,61,62,63,64,65,66,67,68,69,70]. One of the most extensive and successful surveillance programs of PML was conducted by a research group in France. Joly et al. identified 584 patients with PML over the past decade in a nationwide population-based cohort study using the French national health insurance system database and demonstrated the characteristics of PML in France [58]. However, collecting information on PML patients from public databases in Japan remains challenging.

The polymerase chain reaction (PCR) detection of JCV DNA in the cerebrospinal fluid (CSF) is a minimally invasive and reliable indicator of PML diagnosis [71]. Quantitative real-time PCR testing for JCV in the CSF has been performed at the Department of Virology 1 of the National Institute of Infectious Diseases (NIID), Tokyo, Japan, since the fiscal year (FY) 2007 (from April 2007 to March 2008) [72,73,74]. This PCR testing is part of a nationwide surveillance study on PML of the Research Committee of Prion Disease and Slow Virus Infection, which is supported by Health and Labour Sciences Research Grants from the Ministry of Health, Labour and Welfare, Japan. During PCR testing for JCV in the CSF, clinical information of patients suspected of having PML is collected from physicians using standardized questionnaires, and the incidence and characteristics of PML are analyzed (hereafter, “laboratory surveillance”). The prospective laboratory surveillance for PML has been ongoing without interruption and continues. Previously, the results of the study were reported for the start-up period from FY2007 to FY2010 (4 years from April 2007 to March 2011) [72]. Additionally, since FY2016, an ultrasensitive PCR assay capable of detecting at least 10 copies of JCV DNA per mL of CSF has been implemented for the laboratory surveillance of PML [75].

In this study, we conducted a comprehensive analysis of the characteristics of 1537 patients who underwent PCR testing for JCV in the CSF at the NIID, including 288 patients who tested positive, to clarify the overall profile of PML in Japan over the past 10 years (FY2011–2020).

## 2. Materials and Methods

### 2.1. CSF Specimens and Clinical Data

This study was approved by the Medical Research Ethics Committee for the Use of Human Subjects at the NIID (approval nos. 1248 and 1317) and was conducted in accordance with the ethical standards of the Declaration of Helsinki. Written informed consent was obtained from either the patients or their authorized representatives. An Internet-based support system for CSF-JCV testing at the NIID has been described previously [72,75,76]. We received requests for CSF-JCV testing from the patients’ physicians for the diagnosis of PML nationwide, primarily through two websites (available in Japanese only): [http://prion.umin.jp/pml/virus.html] and [https://www.niid.go.jp/niid/ja/from-vir1/9458-vir1-div3.html] (accessed on 3 March 2023). CSF samples collected from patients suspected of PML based on neurological findings and/or magnetic resonance imaging (MRI) patterns were sent to the Department of Virology 1 of the NIID for routine real-time PCR testing for JCV without charge to patients, except for shipping costs. At the time of PCR testing for CSF-JCV, anonymous patient information, such as age, sex, underlying disease, clinical findings, and previous treatment history, was collected by the treating physicians using standardized questionnaires.

### 2.2. Real-Time PCR Testing for JCV in the CSF

From FY2011 to FY2015, the standard PCR assay described previously [72,77,78] was used to quantitatively detect JCV in CSF samples. Total DNAs were extracted from the CSF samples using a QIAamp DNA Blood Mini Kit (Qiagen, Venlo, Netherlands) or a QIAamp Viral RNA Mini Kit (Qiagen) and were subjected to a real-time PCR assay targeting the JCV large T gene. This standard PCR test had a lower detection limit of 200 copies of JCV per mL of CSF [77]. Subsequently, in FY2016–2020, an ultrasensitive PCR assay was introduced as described previously [75], which improved the detection efficiency by highly concentrating JCV DNA during nucleic acid extraction. This allowed for PCR testing with a detection limit of 10 copies/mL by concentrating viral particles in CSF samples with ultrafiltration. The ultrasensitive PCR assay for CSF-JCV was performed according to the procedures described in a previous paper [75], with some modifications. In this study, the Roche LightCycler 96 System (Roche Diagnostics, Mannheim, Germany) was used for real-time PCR instead of the older generation of glass capillary-type instruments. PCR cycles were set according to the manufacturer’s protocol, and the annealing temperature was the same as described before [77]. Additionally, a pUC19 plasmid containing the EcoRI fragment of the JCV Mad-1 strain genome (hereinafter, pMad-1-cl3) was used as standard DNA instead of a BamHI fragment lacking part of the end of the T gene. The PCR detection of JCV in the CSF with a lower limit of 20 copies/mL was routinely performed for ultrasensitive testing. An ultrasensitive JCV test with a detection limit of 10 copies/mL, including the additional step of concentrating JCV particles in the CSF by ultrafiltration, was performed in patients with multiple sclerosis (MS) or when a weak signal was observed in other cases.

### 2.3. Statistical Analysis

The correlation between the numbers of the study population and patients with JCV-positive CSF in the national distribution was evaluated using Spearman’s rank correlation coefficients. In addition, we performed a hierarchical cluster analysis of the regional proportions of patients with JCV-positive CSF using the Ward method. The proportions of patients with JCV-positive CSF and the detection rate of JCV in the specimens in each group were compared using the two-tailed Fisher’s exact test. The copy numbers of the detected JCVs in each group were statistically compared using the Wilcoxon test. All *p*-values < 0.05 were considered statistically significant. The overall incidence rates and 95% confidence interval (CI) of PML cases were calculated based on general population estimates published by the Statistics Bureau of the Ministry of Internal Affairs and Communications of Japan [79].

## 3. Results

### 3.1. Long-Term Performance of Laboratory Surveillance of PML Using PCR Testing for JCV in the CSF

First, we evaluated the performance of the nationwide laboratory surveillance for PML based on PCR testing for JCV in the CSF, which has been ongoing since FY2007. PCR examinations of 2594 specimens, including follow-up tests, were conducted up to FY2020. The number of tests gradually increased from FY2007 to FY2010, which was the start-up period that was previously reported [72]. From FY2011 to FY2020, which was our study period, we observed that 177–254 tests were conducted annually, except for FY2016 (Figure 1A). During the start-up period (FY2007–2010), new requests for JCV testing were frequently made through the website; however, in subsequent periods of this study, 60%–80% of the requests came from the departments of medical institutions that had previously performed testing (Figure 1B). More than 70% of the physicians who requested CSF-JCV testing were from the department of neurology, while only a small percentage of requests were received from other departments (Figure 1C). The number of newly tested individuals reached a peak in FY2014 and declined through FY2016, when the real-time PCR testing for JCV in the CSF became widely available in commercial laboratories. Since then, the number has recovered to approximately 150 annually (Figure 1D). From FY2007 to FY2018, the number of new patients with JCV-positive CSF has been increasing, with at least 30 patients identified annually over the past 5 years (Figure 1E). The percentage of patients with JCV-positive CSF among newly tested patients remained between 10% and 15% from FY2007 to FY2015 but increased sharply in FY2016 and remained high (Figure 1F). These results indicate that the CSF-JCV testing at the NIID was widely recognized during the study period (FY2011–2020), particularly by neurologists, and that the frequency of identifying patients with JCV-positive CSF has been increasing since FY2016.

### 3.2. Geographic Distribution of the Study Population and Patients with JCV-Positive CSF

During the study period (FY2011–2020), CSF-JCV testing was requested by physicians at hospitals in all 47 prefectures of Japan for 1537 new cases suspected of having PML (Figure 2A). The number of requests for CSF-JCV testing showed a pattern similar to the distribution of the Japanese population, with more requests (≥80 cases) received from urban areas, such as Tokyo, Osaka, Kanagawa, Aichi, and Hokkaido (especially Sapporo City). Of these cases, 288 patients (18.7%) had JCV-positive CSF. The geographic distribution of individuals with JCV-positive CSF showed a similar pattern to that of the study population (Figure 2B), and a statistical correlation was found between the two groups (Spearman’s rank correlation coefficient: 0.760, *p* < 0.001). Hierarchical cluster analysis of the percentage of patients with JCV-positive CSF in each prefecture revealed five clusters (Figure 2C). Plotting the trends of these clusters on a map of Japan revealed higher percentages of positive cases in northern coastal prefectures (Figure 2D). Among these clusters, the prefecture with the highest positivity rate (cluster 1) was Akita. The prefectures in cluster 2 were, from east to west, Hokkaido, Niigata, Toyama, Mie, Shimane, Yamaguchi, Saga, Nagasaki, and Okinawa. Major metropolitan areas with a large number of subjects, the Tokyo, Osaka, Kanagawa, and Aichi prefectures were classified as clusters 3 or 4, with relatively low positivity rates. These results suggest that the number of patients with PML correlates with the number of suspected cases, and that the proportion of positive cases is unevenly distributed among regions in Japan.

### 3.3. Overview of Patients with JCV-Positive CSF

During the study period (FY2011–2020), patients with JCV-positive CSF were grouped based on their underlying conditions (Figure 3). Of the 288 positive patients, 19.8% had HIV infection/acquired immunodeficiency syndrome (AIDS), and 43.9% of them were already receiving combined antiretroviral therapy. Additionally, 27.1% and 26.0% of patients with JCV-positive CSF experienced hematological and autoimmune disorders, respectively. The proportions of patients with JCV-positive CSF with a history of solid-organ transplantation, solid tumors, and other diseases accounted for 4.2%, 2.8%, and 13.9%, respectively. Moreover, 6.3% of the patients had no clinically apparent underlying disease identified by their treating physicians despite having JCV-positive CSF. The age and sex distributions of the patients with JCV-positive CSF are shown in Figure 4. When all 288 positive cases were combined, the age range was broadly distributed, with peaks in the 60s, and median ages of 60.5 and 65.0 years for males and females, respectively (Figure 4A). Although there was a higher proportion of males in patients up to their 40s, the male-to-female ratio among all patients with JCV-positive CSF was nearly equal at 1.1 to 1 (152 male patients). However, when the categories of underlying conditions separated the distribution, the majority of individuals with JCV-positive CSF were males in their 30s and 40s for HIV infection (Figure 4B), males and females in their 60s and 70s for hematological disorders (Figure 4C), females in their 50s to 70s for autoimmune disorders (Figure 4D), and males and females in their 60s for other diseases (Figure 4E). Individuals with JCV-positive CSF in the absence of a clinically recognizable underlying disease as a risk factor for PML were primarily males and females in their 80s (Figure 4F). The age distribution and sex ratio patterns of patients with JCV-positive CSF were similar to those of the tested population (Appendix A). These results suggest that PML occurs with various predisposing factors in Japan, and its characteristics differ according to the categories of underlying conditions.

### 3.4. Categories of Underlying Conditions of Patients with JCV-Positive CSF

To evaluate the risk of PML in the study population with different underlying conditions, laboratory surveillance was conducted to collect standardized clinical information for suspected cases, regardless of the CSF-JCV test results. The next set of analyses aimed to determine the JCV detection rate in the CSF (in the study population) and the types of underlying diseases. From FY2011–2020, the proportion of patients with JCV-positive CSF in each category ranged from 22.7% to 27.1%, except for those with solid tumors (8.9%) and those without apparent underlying diseases (4.7%) (Table 1). Notably, the percentages of patients with JCV-positive CSF in various underlying disease categories, excluding those with organ transplantation history or solid tumors, were significantly higher in the second 5 years (FY2016–2020), when ultrasensitive PCR testing was implemented, compared with the first 5 years (FY2011–2015), when standard PCR testing was used. The overall incidence rates of laboratory-identified PML per 100,000 person-years in the first (FY2011–2015) and second (FY2016–2020) periods were 0.016 (95% CI, 0.013–0.020) and 0.029 (95% CI, 0.025–0.034), respectively. These results suggest that PML was more effectively diagnosed in clinical practice during the second 5 years than the first 5 years.

### 3.5. Patients with JCV-Positive CSF and Hematological Disorders

Table 2 displays the percentages of patients with JCV-positive CSF and the types of hematological disorders identified during the study period. JCV was found in 78 out of 305 patients, spanning a wide range of 25 disease types. For each disease type, the number of patients was small, and no statistically significant differences in the proportion of patients with JCV-positive CSF were demonstrated between the first (FY2011–2015) and second (FY2016–2020) periods. During the study period FY2011–2020, 19.2% of patients with JCV-positive CSF had diffuse large B-cell lymphoma (DLBCL). Patients with various types of non-Hodgkin lymphoma, including DLBCL, accounted for 53.8% (42/78) of patients with JCV-positive CSF in this category. JCV was also detected in the CSF of patients with multiple myeloma, leukemia, and other hematological diseases. Although the number of patients with JCV-positive CSF was small, the positivity rates exceeded 40% for patients with 13 of 25 (52%) diseases. These findings suggest that various hematological disorders predispose Japanese patients to PML, and that more than half of the patients with PML in this category had non-Hodgkin lymphoma.

### 3.6. Patients with JCV-Positive CSF and Autoimmune Disorders

The proportions of patients with JCV-positive CSF and autoimmune disorders (75/331 patients) and their disease types are shown in Table 3. In this category, JCV was detected in the CSF samples of patients with 15 different diseases. When comparing the first half (FY2011–2015) to the second half (FY2016–2020) of the study period, no statistically significant difference was found in the percentage of patients with JCV-positive CSF for each disease type. However, the number of patients with JCV-positive CSF in the second period was higher than that in the first period among patients with major PML-predisposing diseases, such as systemic lupus erythematosus (SLE), rheumatoid arthritis (RA), and MS. Of the patients with JCV-positive CSF during the study period FY2011–2020, 45.3% and 13.3% had SLE and RA, respectively. CSF-JCV-positive cases for other disease types were sporadic, accounting for <10% each in this category. However, the positivity rates of CSF-JCV for the entire period were ≥50% for 6 of the 13 diseases. In addition, patients with JCV-positive CSF and several autoimmune diseases were found to have various complications. Data for these cases are summarized in Table 4. With the exception of patients with MS and microscopic polyangiitis, individuals with JCV-positive CSF and autoimmune disorders were identified both with and without complications. These findings suggest that SLE is the predominant type of autoimmune disease underlying PML in Japan, and that PML may develop with or without complications.

### 3.7. Patients with JCV-Positive CSF and Other Underlying Diseases

The percentages and underlying conditions of patients with JCV-positive CSF and a history of organ transplantation, solid tumors, and other diseases (60/308 patients) are listed in Table 5. Owing to the small number of patients in each group, pooled data are tabulated for the entire period. In patients with a history of organ transplantation, 75% of the patients with JCV-positive CSF had received kidney transplantation, indicating a high positivity rate of 28.1% in the tested population. In patients with JCV-positive CSF and solid tumors, the lesions were found in various organs without a distinct pattern. In the groups with the remaining background diseases, 40 patients with JCV-positive CSF were identified. Of these, 16 (40%) had chronic kidney disease, mainly renal failure, whereas patients with JCV-positive CSF and with lung, hepatic, or infectious diseases were rare. In patients with primary immunodeficiency syndromes, the number of patients with JCV-positive CSF was small (*n* = 6). However, all patients with Good syndrome or combined immunodeficiency had JCV-positive CSF. In the group having other diseases, seven patients with sarcoidosis were JCV-positive in the CSF. The positivity rate of JCV in the CSF among patients with sarcoidosis was as high as 46.7%. These results suggest that kidney transplantation, chronic kidney disease, primary immunodeficiency syndrome, and sarcoidosis as well as HIV infection, hematological disorders, and autoimmune disorders are PML-predisposing factors in Japan.

### 3.8. Copy Numbers of JCV in the CSF

The final set of analyses aimed to examine the JCV copy numbers in the CSF during the initial PCR testing of positive cases. The JCV loads in the CSF specimens were widely distributed with a median value of 6.4 × 10^4^ copies/mL during the first 5 years (FY2011–2015), when the standard PCR testing was implemented (Figure 5A, left). During this period, the median JCV levels in patients with HIV infection, hematological disorders, autoimmune disorders, and other diseases were 2.8 × 10^5^, 2.0 × 10^5^, 1.2 × 10^4^, and 1.2 × 10^4^ copies/mL, respectively (Figure 5B–E, left). A broad distribution of JCV copies was also observed during the second 5 years (FY2016–2020), when ultrasensitive PCR testing was introduced (Figure 5A, right). However, the median JCV load in the second period was 2.2 × 10^3^ copies/mL, with a significant number of cases (*n* = 36/186; 19.4%) having low JCV loads of <200 copies/mL. The median JCV copy numbers were in the same order of magnitude (from 1.5 × 10^3^ to 4.0 × 10^3^ copies/mL) in each underlying disease group during the second study period (Figure 5B–F, right). For the total population and groups with HIV infection, hematological disorders, and autoimmune disorders, the JCV levels in the CSF in the second period were statistically significantly lower than those in the first period (Figure 5A–D). These findings suggest that physicians had a better understanding of the risk of PML during the second period, and the ultrasensitive PCR testing in the early stage of the disease detected fewer copy ranges of JCV in the CSF.

## 4. Discussion

In this study, we conducted nationwide laboratory surveillance in Japan for 10 years to analyze the overall picture of PML, a rare disease for which reporting is often not mandatory, except for adverse drug events; thus, assessing its incidence trends in real time is difficult. The ongoing laboratory surveillance in Japan is based on a strategy to identify patients suspected of having PML and predisposing conditions by consolidating CSF-JCV testing in a national laboratory (NIID). However, during the initial four-year start-up period (FY2007–2010), it was difficult to assert that this system had become widespread throughout Japan, mainly due to the simultaneous web-based notification and implementation of CSF-JCV testing at the NIID. Nevertheless, over the next decade, there was an increase in repeat requests from hospital departments with a history of CSF-JCV testing at the NIID, promoting more stable surveillance. In a nationwide population-based cohort study using a public database, the incidence rate of 0.11 per 100,000 person-years was reported for PML in France from 2010 to 2017 [58]. In this study, the incidence rate of PML in Japan was 0.029 per 100,000 person-years, even with an improved surveillance system during the last five years (FY2016–2020). This relatively low incidence of PML in Japan may be attributed to patients with PML who were not tested at the NIID or cases where CSF-JCV could not be detected despite PML development. Alternatively, the relatively small number of individuals with major predisposing factors for PML, such as HIV infection and MS, may account for the low incidence of PML in Japan, as described below.

The hierarchical cluster analysis showed that prefectures with higher CSF-JCV-positive rates among the tested population were mainly distributed along the northern coasts facing the Sea of Japan. However, the reasons for this trend are unknown and require further analysis. Although facilities capable of providing advanced medical care have been established in every prefecture in Japan, the number of hospitals and the frequency of neurology departments tend to be higher in metropolitan areas. Therefore, it is difficult to believe that the level of medical care in neurology is directly reflected in the positivity rate. As JCV is widespread among healthy individuals [15], the regional bias in the infection rate of this virus will unlikely reflect the positivity for JCV in the CSF. Although several sub-lineages of JCV archetypes are circulating in Japan [15], no significant difference in the reactivation or pathogenicity of these viruses was found. Contrastingly, it is difficult to say that many patients with underlying PML-predisposing diseases are found in prefectures with a high CSF-JCV-positivity rate. Compared with other regions (clusters 3–5), the positive rates in these regions (clusters 1 and 2) were statistically significantly higher in patients with HIV infection, hematological disorders, and other diseases (Appendix A). However, the regions with the highest number of people with HIV infection in Japan are metropolitan areas, such as Tokyo, Osaka, and Fukuoka [80], which do not include prefectures with high positive rates for CSF-JCV.

As different underlying conditions and therapeutic interventions predispose patients to PML development, the incidence of PML is likely to vary between countries. Therefore, a detailed list of these predisposing factors and the proportion of patients with PML among suspected cases could aid in clinical management. The age group and sex patterns of the individuals tested and those with JCV-positive CSF in each disease category were comparable, suggesting that the characteristics of predisposing conditions are reflected in PML development. In contrast to the United States and France, where nearly half of PML cases are associated with HIV infection [2,58,81], the proportion of HIV-associated PML cases in Japan is low, approximately 20%, as revealed in this study. The relatively low rate of HIV-associated PML in Japan is consistent with the low prevalence of HIV infection in this country [82].

After the start-up period (FY2007–2010) [72], numerous patients with JCV-positive CSF were found in the group with hematological disorders during the 10-year study period. More than half of the patients with JCV-positive CSF in this group had non-Hodgkin lymphoma. However, other very diverse predisposing factors were also implicated in PML. In addition, the literature search did not identify any case reports of PML associated with essential thrombocythemia or thyroid lymphoma. The data in this study indicate no statistically significant increase in the incidence of PML in the first (FY2011–2015) and second (FY2016–2020) study periods among patients with hematological disorders, partly because of the small number of patients with JCV-positive CSF who had each disease. However, one patient with JCV-positive CSF had multiple myeloma in the first period, compared with eight in the second period, suggesting that PML may increase or continue to occur in Japan over time in the context of multiple myeloma. However, a more detailed analysis of PML in the context of hematologic diseases, including treatment history, is needed.

In contrast to the 4-year start-up period, during which only three patients with JCV-positive CSF were found to have autoimmune disorders [72], 75 patients with JCV-positive CSF were identified in this underlying disease category during the 10-year study period. The reasons for the recent increase in identified PML cases with autoimmune disorders are not fully understood. Approximately 45% of patients with JCV-positive CSF in the autoimmune disease group had SLE, and no major changes, such as the introduction of newly developed drugs, have been observed in managing this disease. Furthermore, although there has been a rise in fingolimod-associated PML cases [83], the number of patients with JCV-positive CSF receiving disease-modifying therapy (DMT) for MS was not very high, with an underlying disease rate of <10% in this category. This observation could reflect the low prevalence of MS and the delayed introduction of new treatments in Japan compared with the United States and Europe [84]. Thus, while DMT-associated PML is a serious concern, it is unlikely to be directly responsible for the significant increase in PML cases with autoimmune disorders in Japan. Nevertheless, increased awareness of DMT-associated PML may have led to a recognition of the importance of considering PML in patients with various autoimmune disorders in Japan. Furthermore, a PubMed search did not reveal any case reports of PML in patients with underlying diseases such as eosinophilic granulomatosis with polyangiitis (previously known as Churg–Strauss syndrome), pemphigus foliaceus, polymyalgia rheumatica, or myelin oligodendrocyte glycoprotein antibody-associated disease. PML surveillance is important from a pharmacovigilance perspective because the use of newly developed drugs and the expansion of indications for existing agents are not uncommon in the treatment of autoimmune diseases.

In this study, laboratory surveillance identified 16 patients with JCV-positive CSF who had chronic kidney disease. In addition, cases of PML in renal failure have been reported [85]. Notably, a non-negligible number of patients with JCV-positive CSF at various stages of chronic kidney disease were observed in this study. Thus, these data demonstrate that chronic kidney disease is a predisposing factor for PML in Japan. In addition, PML rarely occurs in patients with no apparent underlying diseases or with minimal immunosuppression [70,85,86]. Sriwastava et al. conducted a thorough review of the literature on PML in immunocompetent patients and identified 21 published case reports to date, in addition to their own case, with a median patient age of 60 years (range 29–83, SD = 14.0) [70]. In this study, laboratory surveillance identified 18 patients with JCV-positive CSF who had no clinically apparent underlying disease in only one country. Approximately 89% of these patients were seen during the second study period, when ultrasensitive testing was introduced. The patients with JCV-positive CSF had a median age of 77.5 years (range 50–89, SD = 10.3), suggesting that older age may be a predisposing factor for PML in Japan. The exact pathogenesis of PML associated with aging is currently unclear. Although these patients have not been extensively analyzed and have not been clinically diagnosed with an underlying condition, such as idiopathic CD4-positive T lymphocytopenia, it is possible that their T-cell immunity may be compromised due to aging, which could lead to PML. Japan has one of the highest life expectancies globally [87], and it is important to increase clinical awareness of PML among older individuals.

In the second study period, the ultrasensitive PCR testing for JCV in the CSF was introduced into laboratory surveillance to address PML associated with DMT for MS, which often has low viral copy numbers in CSF samples [88]. This large-scale study demonstrated that ultrasensitive JCV testing contributes to PML diagnosis in patients with various predisposing factors and DMT. However, the marked increase in the percentage of CSF-JCV-positive cases among participants in FY2016 and beyond must take into account the increased attention given to PML by physicians in clinical practice in Japan. Additionally, the positivity rates for JCV in the CSF may have been influenced by standard real-time PCR testing in commercial laboratories, which became widespread during this period, as well as ultrasensitive JCV testing. Essentially, the number of patients tested in FY2016 decreased, followed by an increase in the number of patients with JCV-positive CSF. The commercial JCV tests likely screened a larger population, followed by the requests for ultrasensitive JCV testing at the NIID for confirmation and follow-up, resulting in an increase in the number of identifiable PML cases.

Apart from the diagnosis based on pathological findings of brain tissue, PML is clinically diagnosed based on the patient’s neurological symptoms and MRI findings as well as the detection of JCV in the CSF [71]. According to information provided by the physician during follow-up testing, nearly all patients with JCV-positive CSF at the NIID were diagnosed with PML. However, in other cases, no follow-up testing was performed at the NIID after a positive result, and no additional clinical information is available as to whether individuals with JCV-positive CSF were diagnosed with PML or not. While laboratory surveillance using CSF-JCV testing can be the first line for detecting PML and suspected cases, analyzing detailed clinical presentations and MRI data, tracking prognosis over time, and comparing test results performed at a later date in other laboratories are challenging. For these reasons, with the patient’s consent, individuals suspected of PML who have undergone CSF-JCV testing at the NIID are registered in the Research Committee of Prion Disease and Slow Virus Infection, regardless of the PCR results. A multifaceted and integrated analysis of suspected PML cases is then performed by a research team consisting of neurologists, radiologists, pathologists, epidemiologists, and virologists in Japan. The findings of this analysis will be published in the future and will serve as a crucial source of information for PML diagnosis and management.

## 5. Conclusions

Nationwide laboratory surveillance using PCR testing for JCV in CSF samples over the past decade has provided valuable insights into the overall profile of PML in Japan. The data analyzed in this study will not only aid in the diagnosis of PML, but also in understanding the risk of PML associated with underlying diseases and informing their treatment.

## Figures and Tables

**Figure 1 viruses-15-00968-f001:**
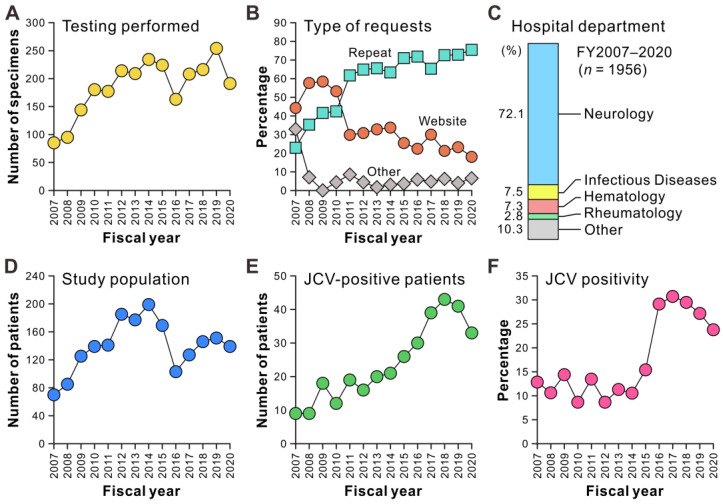
Track record of laboratory surveillance of PML based on PCR testing for JCV in CSF samples. (**A**) Number of CSF-JCV testing performed per year during the 4-year start-up period (FY2007–2010) and the study period (FY2011–2020). The result includes follow-up JCV testing after PML diagnosis. (**B**) Types of requests for CSF-JCV testing from medical departments to the NIID. The requests for CSF-JCV testing for new suspected PML cases in each fiscal year, excluding follow-up testing after PML diagnosis, were classified into three categories: direct contact through the websites, repeat requests from departments that had previously performed the test, and others. (**C**) Hospital departments of physicians who submitted requests for CSF-JCV testing to the NIID. (**D**) Number of new patients with suspected PML tested for CSF-JCV in each fiscal year. (**E**) Number of new patients with JCV-positive CSF in each fiscal year. (**F**) Proportion of patients with JCV-positive CSF among newly tested individuals in each fiscal year.

**Figure 2 viruses-15-00968-f002:**
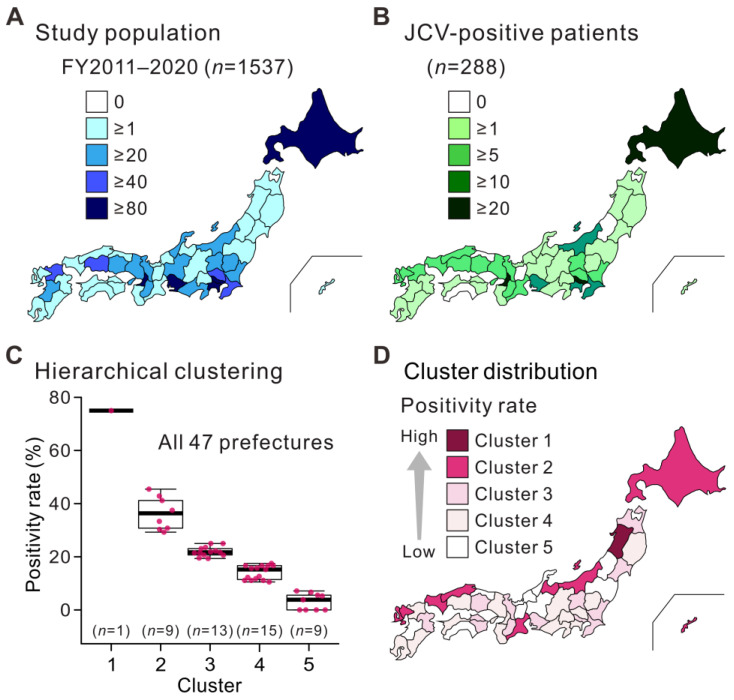
Geographic distribution patterns of individuals tested for JCV in the CSF. All individuals tested (**A**) and patients with JCV-positive CSF (**B**) during the study period (FY2011–2020) are shown. Each prefecture in Japan is color-coded according to the number of individuals. (**C**) The hierarchical cluster analysis of the proportion of patients with JCV-positive CSF among individuals in each prefecture. Japanese prefectures were divided into five clusters according to the CSF-JCV-positivity rates. The dots indicate the positivity rates in each prefecture. The thick horizontal line within each box is the median, and the lower and upper boundaries are the 25th and 75th percentiles, respectively. Vertical whiskers extend to minimum and maximum values. (**D**) Distribution of hierarchical clusters based on CSF-JCV-positivity rates in Japan. Each prefecture is color-coded according to cluster type.

**Figure 3 viruses-15-00968-f003:**
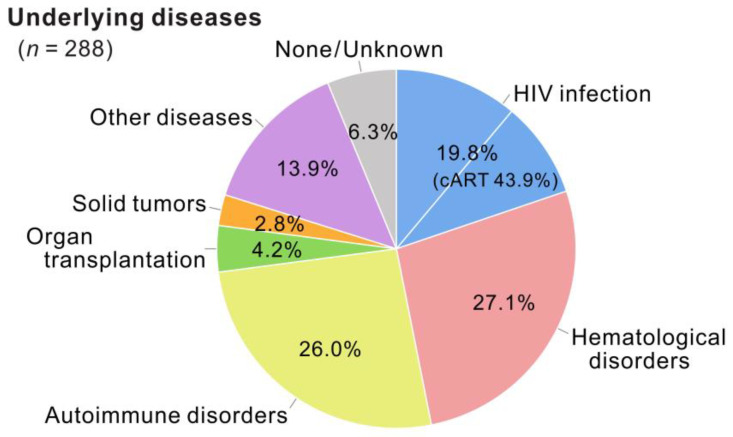
Underlying disease categories of patients with JCV-positive CSF during the study period. The underlying conditions of patients with JCV-positive CSF identified in 10 years (FY2011–2020) are classified into seven categories: HIV infection (*n* = 57), hematological disorders (*n* = 78), autoimmune disorders (*n* = 75), organ transplantation (*n* = 12), solid tumors (*n* = 8), other diseases (*n* = 40), and none/unknown (*n* = 18). Percentages for each category are shown. cART, combined antiretroviral therapy. All percentages are rounded to two decimal places.

**Figure 4 viruses-15-00968-f004:**
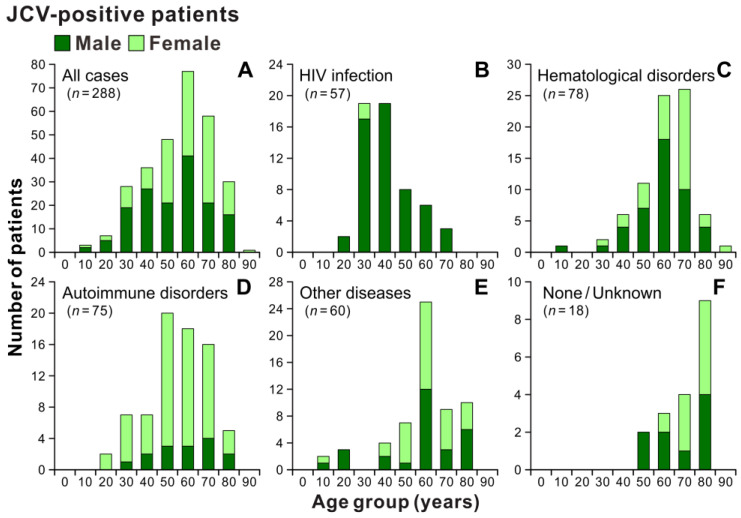
Age and sex distribution patterns of patients with JCV-positive CSF. Patients with JCV-positive CSF during the study period (**A**) were divided into five groups based on underlying conditions: HIV infection (**B**), hematologic disorders (**C**), autoimmune disorders (**D**), other diseases (**E**), and none/unknown (**F**). The “other diseases” group includes patients with a history of organ transplantation, solid tumors, or comorbidities of multiple underlying diseases. The vertical axes show the number of patients, and the numbers below the bars indicate the age groups by decade.

**Figure 5 viruses-15-00968-f005:**
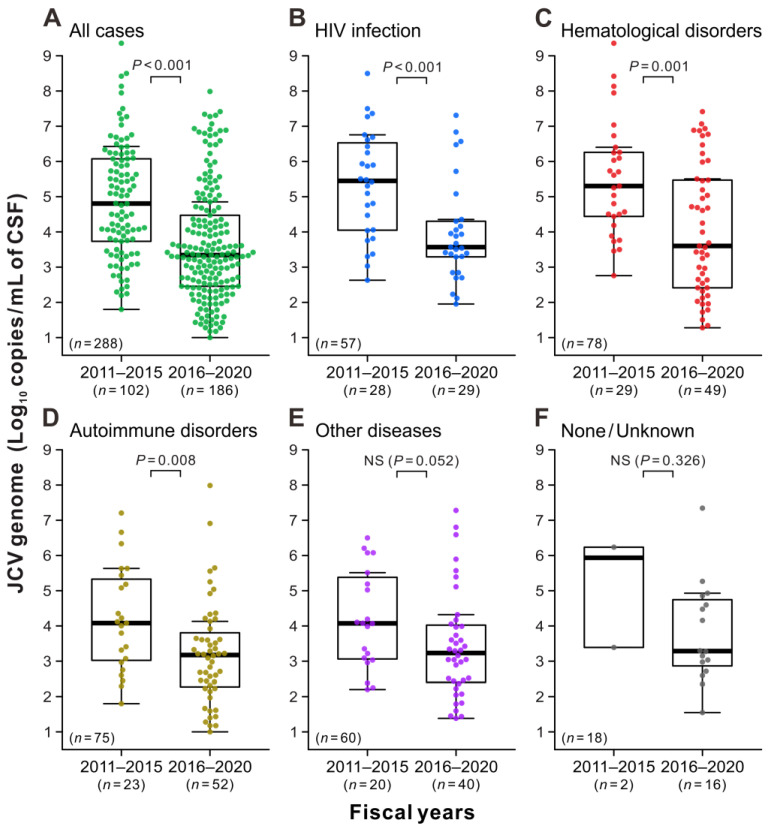
Viral DNA levels in patients with JCV-positive CSF. Patients with JCV-positive CSF during the study period (**A**) were classified into five groups based on underlying conditions: HIV infection (**B**), hematologic disorders (**C**), autoimmune disorders (**D**), other diseases (**E**), and none/unknown (**F**). The copy numbers of the JCV genome in the initial CSF testing that presented positive reactions were indicated using the combinations of beeswarm plots and box-and-whisker plots. In each panel, the data are divided into the first period, when the standard PCR testing was used (FY2011–2015, **left**), and the second period, when the ultrasensitive PCR testing was implemented (FY2016–2020, **right**). The vertical axes show the logarithm of the JCV loads measured by quantitative PCR assay. The thick horizontal line within each box is the median, and the lower and upper limits are the 25th and 75th percentiles, respectively. Vertical whiskers extend to 1.5 times the interquartile range of the data. “NS” indicates no significant difference between the two groups.

**Table 1 viruses-15-00968-t001:** Underlying disease categories and detection rates in patients with JCV-positive CSF.

Category	No. (%) of JCV-Positive Cases ^a^						*p*-Value ^c^
	Entire Period ^b^		FY2011–2015		FY2016–2020		
HIV infection	57/210	(27.1)	28/135	(20.7)	29/75	(38.7)	0.006 *
Hematological disorders	78/305	(25.6)	29/162	(17.9)	49/143	(34.3)	0.001 *
Autoimmune disorders	75/331	(22.7)	23/135	(17.0)	52/196	(26.5)	0.045 *
Organ transplantation	12/47	(25.5)	6/30	(20.0)	6/17	(35.3)	0.306
Solid organ tumors	8/90	(8.9)	4/58	(6.9)	4/32	(12.5)	0.447
Other diseases	40/171	(23.4)	10/105	(9.5)	30/66	(45.5)	<0.001 *
None/unknown	18/383	(4.7)	2/246	(0.8)	16/137	(11.7)	<0.001 *
Total	288/1537	(18.7)	102/871	(11.7)	186/666	(27.9)	<0.001 *

Abbreviations: CSF, cerebrospinal fluid; FY, fiscal years; HIV, human immunodeficiency virus; JCV, JC virus. ^a^ The number of JCV-positive cases and positive rates (%) are indicated. ^b^ Entire period indicates FY2011–2020. ^c^ The positive rates of CSF-JCV in FY2011–2015 and FY2016–2020 are statistically analyzed using Fisher’s exact test. Significant differences are indicated by asterisks (*p* < 0.05).

**Table 2 viruses-15-00968-t002:** PCR-based detection of JCV in CSF specimens from patients with hematological disorders.

Underlying Disease ^b^	No. (%) of JCV-Positive Cases ^a^						% of Total ^d^
	FY2011–2015		FY2016–2020		Entire Period ^c^		
Diffuse large B-cell lymphoma	7/26	(26.9)	8/32	(25.0)	15/58	(25.9)	19.2
Multiple myeloma	1/3	(33.3)	8/19	(42.1)	9/22	(40.9)	11.5
Chronic lymphocytic leukemia	3/6	(50.0)	5/6	(83.3)	8/12	(66.7)	10.3
Non-Hodgkin lymphoma (unspecified)	2/20	(10.0)	6/15	(40.0)	8/35	(22.9)	10.3
Mucosa-associated lymphoid tissue lymphoma	3/5	(60.0)	1/2	(50.0)	4/7	(57.1)	5.1
Follicular lymphoma	1/10	(10.0)	3/9	(33.3)	4/19	(21.1)	5.1
Essential thrombocythemia	0/0	− ^e^	3/3	(100)	3/3	(100)	3.8
Acute myeloid leukemia	2/19	(10.5)	1/6	(16.7)	3/25	(12.0)	3.8
Adult T-cell leukemia/lymphoma	2/13	(15.4)	1/12	(8.3)	3/25	(12.0)	3.8
Acute lymphoid leukemia	1/16	(6.3)	2/10	(20.0)	3/26	(11.5)	3.8
Primary macroglobulinemia	2/3	(66.7)	0/0	−	2/3	(66.7)	2.6
Idiopathic CD4 lymphocytopenia	1/2	(50.0)	1/1	(100)	2/3	(66.7)	2.6
Hodgkin lymphoma	0/4	(0)	2/2	(100)	2/6	(33.3)	2.6
Angioimmunoblastic T-cell lymphoma	1/1	(100)	0/0	−	1/1	(100)	1.3
Thyroid lymphoma	1/1	(100)	0/0	−	1/1	(100)	1.3
Small lymphocytic lymphoma	1/1	(100)	0/0	−	1/1	(100)	1.3
Hypereosinophilic syndrome	0/0	−	1/1	(100)	1/1	(100)	1.3
Sézary syndrome	0/0	−	1/1	(100)	1/1	(100)	1.3
Mycosis fungoides	0/0	−	1/1	(100)	1/1	(100)	1.3
Mantle cell lymphoma	0/1	(0)	1/1	(100)	1/2	(50.0)	1.3
Lymphoplasmacytic lymphoma	0/2	(0)	1/1	(100)	1/3	(33.3)	1.3
Aplastic anemia	0/1	(0)	1/2	(50.0)	1/3	(33.3)	1.3
Lymphoblastic lymphoma	0/1	(0)	1/3	(33.3)	1/4	(25.0)	1.3
Chronic myeloid leukemia	1/4	(25.0)	0/3	(0)	1/7	(14.3)	1.3
Myelodysplastic syndrome	0/5	(0)	1/4	(25.0)	1/9	(11.1)	1.3
Other diseases	0/18	(0)	0/9	(0)	0/27	(0)	0
Total	29/162	(17.9)	49/143	(34.3)	78/305	(25.6)	100

Abbreviations: CD, cluster of differentiation; CSF, cerebrospinal fluid; FY, fiscal years; JCV, JC virus. ^a^ Numbers of JCV-positive cases and positive rates (%). ^b^ Underlying diseases are sorted by the number of positive cases for FY2011–2020. If the number of positive cases is the same, then they are ranked by a positive rate. ^c^ Entire period indicates FY2011–2020. ^d^ Percentages of underlying disease among all JCV-positive cases with hematological disorders for the entire period. ^e^ The minus symbol indicates not applicable.

**Table 3 viruses-15-00968-t003:** PCR-based detection of JCV in CSF specimens from patients with autoimmune disorders.

Underlying Disease ^b^	No. (%) of JCV-Positive Cases ^a^						% of Total ^d^
	FY2011–2015		FY2016–2020		Entire Period ^c^		
Systemic lupus erythematosus	13/36	(36.1)	21/37	(56.8)	34/73	(46.6)	45.3
Rheumatoid arthritis	2/31	(6.5)	8/34	(23.5)	10/65	(15.4)	13.3
Multiple sclerosis	1/9	(11.1)	6/58	(10.3)	7/67	(10.4)	9.3
Dermatomyositis	1/2	(50.0)	3/4	(75.0)	4/6	(66.7)	5.3
Autoimmune hemolytic anemia	0/0	− ^e^	3/3	(100)	3/3	(100)	4.0
Eosinophilic granulomatosis with polyangiitis	1/2	(50.0)	2/3	(67.0)	3/5	(60.0)	4.0
Sjögren syndrome	1/2	(50.0)	2/4	(50.0)	3/6	(50.0)	4.0
Antineutrophil cytoplasmic antibody- associated vasculitis	1/3	(33.3)	2/6	(33.3)	3/9	(33.3)	4.0
Pemphigus foliaceus	1/1	(100)	1/1	(100)	2/2	(100)	2.7
Autoimmune hepatitis	0/1	(0)	1/1	(100)	1/2	(50.0)	1.3
Systemic scleroderma	0/2	(0)	1/1	(100)	1/3	(33.3)	1.3
Polymyalgia rheumatica	1/3	(33.3)	0/1	(0)	1/4	(25.0)	1.3
Myelin oligodendrocyte glycoprotein antibody-associated disease	0/0	−	1/4	(25.0)	1/4	(25.0)	1.3
Granulomatosis with polyangiitis	1/3	(33.3)	0/2	(0)	1/5	(20.0)	1.3
Microscopic polyangiitis	0/5	(0)	1/2	(50.0)	1/7	(14.3)	1.3
Other diseases	0/35	(0)	0/35	(0)	0/70	(0)	0
Total	23/135	(17.0)	52/196	(26.5)	75/331	(22.7)	100

Abbreviations: CSF, cerebrospinal fluid; FY, fiscal years; JCV, JC virus. ^a^ Numbers of JCV-positive cases and positive rates (%). ^b^ Underlying diseases are sorted by the number of positive cases for FY2011–2020. If the number of positive cases is the same, then they are ranked by a positive rate. ^c^ Entire period indicates FY2011–2020. ^d^ Percentages of underlying disease among all JCV-positive cases with autoimmune disorders for the entire period. ^e^ The minus symbol indicates not applicable.

**Table 4 viruses-15-00968-t004:** Complications of autoimmune disorders in patients with JCV-positive CSF.

Underlying Diseases ^a^	Complications	Entire Period (FY2011–2020)		
		No. (%) of JCV-Positive Cases ^b^		% of Disease ^c^
Systemic lupus erythematosus	None	10/32	(31.3)	29.4
	Lupus nephritis	12/22	(54.5)	35.3
	Antiphospholipid syndrome	4/4	(100)	11.8
	Lupus nephritis, Sjögren syndrome	2/2	(100)	5.9
	Rheumatoid arthritis	2/2	(100)	5.9
	Sjögren syndrome	2/5	(40.0)	5.9
	Non-Hodgkin lymphoma	1/1	(100)	2.9
	Systemic scleroderma, polymyositis	1/1	(100)	2.9
	Others	0/4	(0)	0
Rheumatoid arthritis	None	6/44	(13.6)	60.0
	Mixed connective tissue disease, malignant lymphoma	1/1	(100)	10.0
	Hodgkin lymphoma	1/1	(100)	10.0
	Non-Hodgkin lymphoma	1/5	(20.0)	10.0
	Chronic kidney disease	1/6	(16.7)	10.0
	Others	0/8	(0)	0
Multiple sclerosis	None	7/64	(10.9)	100
	Rheumatoid arthritis, non-Hodgkin lymphoma	0/1	(0)	0
	Hyperthyroidism	0/1	(0)	0
	Behçet disease	0/1	(0)	0
Autoimmune hemolytic anemia	None	2/2	(100)	66.7
	Malignant lymphoma	1/1	(100)	33.3
Sjögren syndrome	None	1/2	(50.0)	33.3
	Non-Hodgkin lymphoma	2/2	(100)	66.7
	Chronic hepatitis C	0/1	(0)	0
	Interstitial pneumonia	0/1	(0)	0
ANCA-associated vasculitis	None	1/6	(16.7)	33.3
	ANCA-associated glomerulonephritis	2/2	(100)	66.7
	IgG4-related disease	0/1	(0)	0
Microscopic polyangiitis	None	0/5	(0)	0
	Chronic kidney disease, interstitial pneumonia	0/1	(0)	0
	Chronic kidney disease	1/1	(100)	100
Total	− ^d^	61/230	(26.5)	−

Abbreviations: ANCA, antineutrophil cytoplasmic antibody; CSF, cerebrospinal fluid; FY, fiscal years; IgG4, immunoglobulin G4; JCV, JC virus; NA, not applicable. ^a^ Autoimmune disorders without complications are excluded. ^b^ Numbers of JCV-positive cases and positive rates (%). ^c^ Percentages of JCV-positive cases in each underlying disease. ^d^ The minus symbol indicates not applicable.

**Table 5 viruses-15-00968-t005:** PCR-based detection of JCV in CSF specimens from patients with organ transplantation, solid cancers, and other diseases.

Category	Subcategory ^a^	Entire Period (FY2011–2020)		
		No. (%) of JCV-Positive Cases ^b^		% of Category ^c^
Organ transplantation	Kidney	9/32	(28.1)	75.0
	Liver	2/12	(16.7)	16.7
	Lung	1/1	(100)	8.3
	Heart	0/2	(0)	0
Solid organ tumors	Liver	2/14	(14.3)	25.0
	Hypopharynx	1/1	(100)	12.5
	Thymus	1/3	(33.3)	12.5
	Esophagus	1/6	(16.7)	12.5
	Stomach	1/7	(14.3)	12.5
	Lung	1/25	(4.0)	12.5
	Others ^d^	1/34	(2.9)	12.5
Brain diseases	− ^e^	0/25	(0)	−
Heart diseases	−	0/5	(0)	−
Lung diseases	Interstitial pneumonia	1/5	(20.0)	100
	Others	0/1	(0)	0
Liver diseases	Hepatitis C-related cirrhosis	2/7	(28.6)	66.7
	Cirrhosis (unspecified)	1/2	(50.0)	33.3
	Others	0/8	(0)	0
Kidney diseases	CKD (renal failure)	10/29	(34.5)	62.5
	CKD (nephrotic syndrome)	1/2	(50.0)	6.3
	CKD (nephrosclerosis)	1/3	(33.3)	6.3
	CKD (unspecified)	1/3	(33.3)	6.3
	CKD (polycystic kidney disease)	1/4	(25.0)	6.3
	CKD (chronic glomerulonephritis)	1/7	(14.3)	6.3
	CKD (diabetic nephropathy)	1/9	(11.1)	6.3
	Others	0/2	(0)	0
Infectious diseases	Non-tuberculous mycobacterial infection	1/1	(100)	100
	Others	0/4	(0)	0
Primary immunodeficiency syndromes	Good syndrome	4/4	(100)	66.7
	Combined immunodeficiency syndrome	1/1	(100)	16.7
	Chronic granulomatous disease	1/3	(33.3)	16.7
	Others	0/4	(0)	0
Other diseases	Sarcoidosis	7/15	(46.7)	53.8
	Comorbidity ^f^	5/9	(55.6)	38.5
	Amyloidosis	1/2	(50.0)	7.7
	Others	0/16	(0)	0
Total	−	60/308	(19.5)	−

Abbreviations: CKD, chronic kidney disease; CSF, cerebrospinal fluid; FY, fiscal years; JCV, JC virus. ^a^ The subcategories within each category are arranged according to the number of CSF-JCV-positive cases for the entire period. If the number of positive cases is the equal, then they are ranked in order of positive rate. ^b^ Numbers of JCV-positive cases and positive rates (%). ^c^ Percentages of subcategories in each category. ^d^ One patient with JCV-positive CSF had germ cell tumor. ^e^ The minus symbol indicates not applicable. ^f^ This subcategory includes cases with multiple underlying conditions whose relationship to each other could not be determined from the information in the questionnaire.

## Data Availability

The analyzed datasets are available in the article and its Appendix A, or are available from the corresponding author upon reasonable request.

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
