# Peer review of "Nationwide Laboratory Surveillance of Progressive Multifocal Leukoencephalopathy in Japan: Fiscal Years 2011–2020"

_viruses, 2023, doi:10.3390/v15040968_

Round 1

Reviewer 1 Report

The review by Nakamichi  and colleagues is of considerable importance for international virology because it brings to the attention of researchers and clinicians a serious and often lethal disease such as PML due to JCV, both of which are much neglected not only as a result of the SARS pandemic -CoV-2.

The work is well organized and provides an accurate picture of the frequency and distribution of PML from a centralized reference laboratory throughout Japan where almost all PCRs are performed for diagnosis and/or confirmation of the presence of JCV DNA in the CSF of suspected or confirmed cases of PML.

The description of the results is very clear and convincing and allows to obtain a series of information such as the frequency per year (starting from 2011 to 2020), the geographical distribution of the cases studied, the underlying diseases that were at the basis of the occurrence of PML, including the distribution by gender.

The authors have proposed a very accurate description of the frequency of positive JCV cases among the various underlying diseases which is very useful to understand how PML is no longer associated only with HIV, but manifests itself in many other diseases or surgical procedures as a result the use of immunosuppressive therapies.

This information is of particular value to clinicians dealing with these diseases who may be led to delay or underestimate the possible search for JCV DNA in the CSF. Very useful in this sense are the accurate lists of autoimmune and hematological diseases that may be the basis for the onset of PML, and the frequency among the different types of organ transplants and solid tumors.

The review is also one of the few that also includes virological follow-up of patients after PML diagnosis. Furthermore, it is very important to demonstrate the importance of using an ultra sensitive PCR in improving the sensitivity and specificity of the detection of JCV DNA in the CSF.

The paper is, as mentioned, well organized and well written, can be read with pleasure and contains a series of important and useful data.

There is only one suggestion for the Authors: they should provide some clarification regarding the geographical distribution, especially for readers who are not familiar with Japan. It would be useful to know if there is a different frequency of hospitals, neurology departments in the various districts, and in particular if it is clear why, as indicated in Figure 2, the cluster distribution (panel D) seems to be different from Panel A and D .

 With this small addition the paper can be published.

Author Response

Response: We would like to express our sincere respect and gratitude to Reviewer 1 for recognizing the value of this research and for making important suggestions. We have modified the Results section of the manuscript to clarify the geographic distribution of the study subjects (lines 225–228) and positivity rates of CSF-JCV (lines 234–239). We added the sentences "Although facilities capable of providing advanced medical care have been established in every prefecture in Japan, the number of hospitals and the frequency of neurology departments tend to be higher in metropolitan areas. Therefore, it is difficult to believe that the level of medical care in neurology is directly reflected in the positivity rate." in the Discussion section of the manuscript (lines 464–468).

Reviewer 2 Report

I really enjoyed reading the review regarding the national laboratory surveillance of PML in Japan. It is very well written and well organised.

It is very encouraging to observe that there is a resurgence in interest in infections other than SARS-CoV-2 which for a long time, have inevitably been overlooked. The review by Nakamichi and colleagues highlights the importance of the introduction of ultra-sensitive PCR assays in 2016, which has improved the laboratory surveillance of PML. Furthermore, the authors underline the importance of virological follow-up of patients who have been diagnosed with PML, which unfortunately is not always done.

It would be desirable to have a picture similar to the one described in this review, for each geographical area in the world, in order to be able to compare the epidemiological situations, deepening them.

The article is publishable as is.

Author Response

Response: We would like to express our profound appreciation to Reviewer 2 for the tremendous encouragement of our work. As an opportunistic infection, PML reflects the underlying disease situation in each country. We agree with Reviewer 2 that studying this disease on a global scale is crucial.
